# Clinicopathologic Characteristics, Treatment Outcomes, and Survival in Thymic Neuroendocrine Neoplasms (t-NEN): A 25-Year Single-Center Experience

**DOI:** 10.3390/cancers17243932

**Published:** 2025-12-09

**Authors:** Aleksandra Piórek, Adam Płużański, Dariusz M. Kowalski, Maciej Krzakowski

**Affiliations:** Department of Lung Cancer and Thoracic Tumors, Maria Sklodowska-Curie National Research Institute of Oncology, 02-781 Warsaw, Polanddariusz.kowalski@nio.gov.pl (D.M.K.); maciej.krzakowski@nio.gov.pl (M.K.)

**Keywords:** thymic neuroendocrine neoplasms, t-NENs, atypical carcinoid, clinical management, surgical resection, systemic therapy, recurrence, prognosis

## Abstract

Thymic neuroendocrine neoplasms (t-NENs) are exceptionally rare tumors, and most current knowledge is derived from small retrospective studies. We analyzed 19 cases treated over a 25-year period at a comprehensive cancer center. The most common histologic subtype was atypical carcinoid, and nearly half of the patients treated with curative intent experienced disease recurrence. The median overall survival (OS) for the entire cohort was 127 months, with significantly longer OS observed in curatively treated patients (170 months) compared to those receiving palliative care (33 months). Despite multimodal therapy, recurrence and disease progression remained common. Our findings highlight the need for more effective treatment strategies and close long-term follow-up.

## 1. Introduction

Thymic neuroendocrine neoplasms (t-NENs) represent a rare and biologically aggressive subset of tumors arising in the anterior mediastinum. They account for less than 5% of all thymic malignancies and under 0.5% of all neuroendocrine neoplasms [1]. According to the 2021 World Health Organization (WHO) classification, t-NENs are categorized into well-differentiated neuroendocrine tumors—typical and atypical carcinoids—and poorly differentiated neuroendocrine carcinomas, including small-cell and large-cell subtypes [2].

Given their exceptionally low incidence, estimated at 0.02–0.18 per 100,000 individuals annually, clinical knowledge on prognostic indicators, treatment strategies, and long-term outcomes is limited. Most available data are derived from retrospective case series or institutional experiences [1,3].

The disease shows a male predominance, accounting for approximately 70–80% of cases, and is typically diagnosed between the fifth and sixth decade of life [4,5,6,7]. A subset of t-NENs occurs in the context of multiple endocrine neoplasia type 1 (MEN1), a hereditary syndrome associated with a poorer prognosis and an increased risk of metastatic spread [4,5,7].

Given the exceptional rarity and biological heterogeneity of t-NENs, comprehensive data on their clinical behavior and optimal treatment strategies remain scarce. The present study aimed to perform a retrospective analysis of the clinicopathological characteristics, treatment modalities, and survival outcomes of patients with t-NENs treated at one of the largest comprehensive cancer centers in Poland over the past 25 years. Additionally, the study sought to identify potential prognostic factors that may influence treatment results and disease course.

## 2. Materials and Methods

A retrospective analysis was conducted on the medical records of 19 adult patients diagnosed with t-NENs who were treated at the National Research Institute of Oncology in Warsaw, Poland, between January 2000 and December 2024. Patients were identified from our institution database. The analysis included adult individuals (≥18 years) with a diagnosis of t-NEN.

Demographic information, clinicopathologic variables (including symptoms at presentation, smoking history, performance status, histological diagnosis, and tumor extent), and treatment modalities were extracted from patient records.

Histological classification was performed according to the 2021 World Health Organization (WHO) guidelines. For cases diagnosed prior to 2021, histopathological classification was retrospectively adjusted to align with the WHO 2021 criteria, without re-review by pathologists. Well-differentiated tumors included typical carcinoid (TC) and atypical carcinoid (AC), while poorly differentiated tumors comprised small-cell carcinoma and large-cell neuroendocrine carcinoma (LCNEC). In cases where the histological subtype could not be clearly determined based on the available documentation, the diagnosis was classified as “neuroendocrine tumor, not otherwise specified (NOS)”. Although the 2021 WHO classification does not include a formal NOS category for thymic NEN, it was employed in this study for descriptive purposes when precise subtype classification was not feasible, but the neuroendocrine nature of the tumor was histologically confirmed.

As no dedicated staging system exists for t-NENs, clinical staging in the study cohort was heterogeneous: some cases were classified using the Masaoka-Koga system, others using the TNM classification developed for thymomas, and in several instances no staging data were available. Due to incomplete documentation—particularly in older cases—retrospective staging was not feasible and thus was not included in the statistical analysis. Given the heterogeneity of the study population, analyses were conducted for the entire cohort as well as stratified by treatment intent (curative vs. palliative).

The rates of radical (R0) resections and the presence of residual disease (R1/R2) were analyzed. Surgical treatment was considered radical (R0) when complete tumor excision with histologically negative margins was achieved. Incomplete resection was defined as the presence of microscopic (R1) or macroscopic (R2) residual disease. Diagnostic biopsy alone was classified as “no surgery.”

Survival rates (OS, DFS, and PFS) were estimated using the Kaplan–Meier method. OS was calculated from the date of pathological diagnosis to the date of death from any cause. Survival curves were generated using the Kaplan–Meier method, and differences between groups were assessed with the log-rank test. A two-sided *p*-value < 0.05 was considered statistically significant (95% confidence interval).

Follow-up was complete and survival data were updated for all 19 patients as of 31 March 2025. No patients were lost to follow-up, and vital status was confirmed in all cases. The study was conducted in accordance with the Declaration of Helsinki and was approved by the Institutional Review Board of the National Research Institute of Oncology.

## 3. Results


**Study Population**


Between 2000 and 2024, 19 adult patients t-NEN were treated at the Maria Sklodowska-Curie National Research Institute of Oncology. The median age at diagnosis was 52 years (range: 24–73), and the majority were male (12 patients, 63.2%). Symptoms were present at diagnosis in 73.7% of patients, with dyspnea and chest pain being the most common. Paraneoplastic syndromes were observed in 5 patients (26.3%), including clinically confirmed Cushing’s syndrome in 3 cases (15.8%).


**Histopathology**


Among the 19 patients included in the study, atypical carcinoid was the most frequently observed histological subtype, diagnosed in 10 cases (52.6%). Large-cell neuroendocrine carcinoma (LCNEC) was identified in 6 patients (31.6%). In 3 cases (15.8%), the available histopathological data were insufficient to assign a specific subtype; these tumors were classified as neuroendocrine tumors, not otherwise specified (NET NOS). While the 2021 WHO classification does not formally recognize a NOS category for thymic NENs, this designation was used descriptively in cases where neuroendocrine differentiation was confirmed, but more precise classification was not feasible.

The distribution of histological subtypes in the study cohort is summarized in Table 1.


**Treatment**


Curative-intent treatment was performed for 9 patients (47.4%), while 10 patients (52.6%) received palliative care. Radical surgery was attempted for 8 patients. Complete resection (R0) was achieved in 2 cases (25%), and microscopic residual disease (R1) was present in 6 patients (75%). Postoperative radiotherapy was administered in 7 patients (36.8%), with total doses ranging from 4600 to 6000 cGy. Three patients (15.8%) received adjuvant chemotherapy based on a cisplatin-etoposide regimen (PE).

In the palliative group, 8 patients (80.0%) received systemic chemotherapy—most commonly with a PE regimen—while 2 patients (20.0%) received best supportive care alone.

Treatment modalities and proportions across the study cohort are summarized in Table 2.


**Recurrence and Progression**


Disease recurrence occurred in 4 of 9 patients (44.4%) treated with curative intent. Of these, one patient received salvage radical-intent treatment, while three were referred to palliative care. Progression was observed in 8 of 10 patients (80%) initially treated with palliative intent. Seven of these patients received further active palliative therapy, including radiotherapy (*n* = 3) and chemotherapy (*n* = 4).


**Survival Outcomes**


For the entire cohort, the median OS was 127 months [95% CI: 33–170]. In the curatively treated group, the median DFS was 90 months [95% CI: 42–90], and the median OS was 170 months [95% CI: 127–170]. Among palliatively treated patients, the median PFS was 11 months [95% CI: 3–105], and the median OS was 33 months [95% CI: 2–65].

Detailed survival outcomes stratified by treatment intent are summarized in Table 3, and Kaplan–Meier survival curves are presented in Figure 1.

## 4. Discussion

Thymic neuroendocrine neoplasms (t-NENs) are exceptionally rare and aggressive malignancies of the anterior mediastinum, as evidenced both by the existing literature and by the findings of our retrospective analysis involving 19 patients treated over a 25-year period. Their incidence is estimated at 0.02–0.18 per 100,000 individuals annually, which results in a limited number of prospective studies and a significant fragmentation of available data [1].

In our cohort, the median age of our patients (52 years) and the predominance of male sex (63%) are consistent with data reported in the literature [4,8]. In the largest series comprising 205 cases of t-NENs, a male predominance of 77% was observed, with a median age of approximately 54 years [9]. Two patients were diagnosed with multiple endocrine neoplasia type 1 (MEN1). Approximately 20–25% of t-NENs occur in patients with MEN1, and thymic tumors themselves account for a significant proportion (up to ~20%) of MEN1-related mortality [4]. Moreover, thymic carcinoids in the context of MEN1 are associated with particularly poor prognosis—they occur almost exclusively in men (often smokers) and are characterized by high lethality [5]. Ferolla et al. (2005) reported seven such cases in MEN1 patients (all male, 85% of whom were smokers) and suggested that prophylactic thymectomy should be considered during parathyroid surgery in MEN1 males to prevent the development of occult thymic tumors [5]. It is also recommended that all patients diagnosed with t-NEN be screened for MEN1 (and conversely, that periodic mediastinal imaging be performed in MEN1 patients) due to the impact of this syndrome on treatment strategy and surveillance [4,5]. Although molecular profiling and germline testing were not routinely performed in our cohort, recent studies highlight the potential role of mutational analysis—both somatic and germline—in the management of thymic neuroendocrine tumors. Identification of MEN1 gene mutations is particularly relevant in the context of hereditary syndromes such as MEN1 [10]. Moreover, molecular markers like PD-L1 or MGMT may help guide therapeutic decisions in the future. PD-L1 expression and immune cell infiltration observed in a subset of t-NENs suggest that these tumors might benefit from immunotherapy [11], while MGMT expression may predict response to temozolomide-based chemotherapy [8]. As access to next-generation sequencing expands, integrating molecular profiling into clinical protocols may enhance individualized treatment approaches in this rare tumor entity.

In our cohort, the most common histologic subtype was atypical carcinoid (52.6%), followed by LCNEC (31.6%); no cases of TC (well-differentiated typical carcinoid) were identified. This predominance of highly aggressive subtypes over low-grade neoplasms is in line with the data reported in the literature. In a large multicenter analysis, AC accounted for approximately 40% of cases, representing the most frequent histology [9]. Typical thymic carcinoids are rare—unlike their pulmonary counterparts, thymic tumors are more frequently of intermediate or high grade. Despite the misleading nomenclature implying indolence, thymic carcinoids have a significantly more aggressive clinical course compared to their pulmonary or gastrointestinal equivalents [2]. Notably, our 25-year dataset did not include a single case of TC, underscoring this biological distinction. Similar findings have been reported elsewhere—for example, Gaur et al. described the absence of low-grade tumors among 12 analyzed t-NENs, classifying 100% of cases as either AC or neuroendocrine carcinoma [12]. This further supports the notion that even histologically “well-differentiated” thymic neoplasms typically exhibit more aggressive clinical behavior than well-differentiated NETs in other organ systems.

Three cases in our study were classified as “neuroendocrine tumor, not otherwise specified” (NET NOS). Although the 2021 WHO classification does not formally include this category for thymic neuroendocrine neoplasms, we adopted the term descriptively in reference to cases where the available documentation confirmed the neuroendocrine nature of the tumor but lacked sufficient detail for definitive histologic subtyping [2]. A similar pragmatic approach has been observed in other published case series, which used labels such as NET NOS, “unclassified NET,” or “non-typable” in contexts of limited histopathological data [4,8,11].

Radical surgical treatment is broadly recognized as a key component in the management of t-NENs with curative intent, and our results appear to support this approach. The median OS in patients who underwent curative treatment was 170 months, compared to 33 months in the palliative group. In our cohort, only 25% of operated patients achieved R0 resection. This finding aligns with results from other centers [6] and reflects the broader challenges noted in international databases [9,12].In the multicenter study by Filosso et al., R0 resection was feasible in only 57% of operated patients [9]. A detailed comparison of treatment characteristics and survival outcomes from retrospective studies is presented in Table 4.

Advanced local stage (Masaoka stage III/IV) and incomplete resection are the most significant adverse prognostic factors related to treatment. Higher clinical stage and non-radical resection have been shown to significantly reduce both survival and time to recurrence [9]. Our data also indicate that nearly half of the patients who underwent curative-intent treatment (radical surgery ± adjuvant chemoradiotherapy) experienced disease recurrence. In our series, the recurrence rate was approximately 44%. This is consistent with findings from other institutions—for example, Yliaska et al. (2022) reported a recurrence rate of 46%, while Cardillo et al. (2012) reported as high as 67% following surgical resection [4,6].

The high rate of recurrence observed despite surgical treatment suggests that adjuvant therapies may be beneficial in selected cases. The literature also supports the role of radiotherapy— postoperative irradiation has been reported to improve local disease control and may be associated with prolonged progression-free survival in some studies [11]. In the SEER database analysis, adjuvant radiotherapy was associated with benefits in patients with more advanced disease, improving both local control and overall survival [16].

Recurrence may present as either locoregional or distant disease. Studies by Zhai et al. (2022) and Cheng et al. (2024) have emphasized the high incidence of distant metastases, particularly to the bone, reported in up to 23% of cases [8,11]. These findings underscore the necessity for strict postoperative surveillance and early detection of disseminated disease. We recommend regular follow-up imaging—computed tomography (CT) or magnetic resonance imaging (MRI) of the chest—and, when clinically indicated, targeted diagnostic workup for bone metastases. In selected cases, the use of radiolabeled somatostatin analogs (e.g., receptor scintigraphy or PET imaging) may be helpful for detecting occult hormonally active foci [1].

In the context of systemic therapy, our standard regimen consisted of cisplatin combined with etoposide—a protocol adapted from the treatment of small cell lung carcinoma (SCLC). This approach reflects common practice in other oncology centers as well, given the lack of dedicated prospective clinical trials for t-NENs. In the study by Ferolla et al. (2005), first-line treatment was also based on a platinum-etoposide combination, frequently combined with radiotherapy; however, the authors noted a high rate of recurrence despite initial responses to treatment [5]. Similarly, Cheng et al. (2024) reported limited efficacy of chemotherapy, with a median PFS of 10 months and a low disease control rate [8]. In our palliative group, disease progression was observed in 80% of patients treated with cisplatin and etoposide, further confirming the limited effectiveness of this regimen in advanced t-NENs. The median PFS was approximately 11 months, which is comparable to outcomes reported for first-line chemotherapy in advanced pulmonary neuroendocrine carcinomas [16]. Other studies have indicated that more than half of patients with extensive metastatic disease fail to achieve 2-year survival [9,11].

Due to poor outcomes associated with standard platinum-based chemotherapy, alternative treatment options are being explored for patients with advanced t-NENs. In our series, systemic therapies in the palliative setting yielded limited results, with disease progression in 80% of cases and a median PFS of 11 months. While data from small studies suggest potential benefit from temozolomide-based regimens—particularly in tumors expressing MGMT—and immune checkpoint inhibitors in PD-L1–positive cases, these approaches remain investigational [1,2,8,17]. Further research is needed to clarify their role in t-NENs.

Despite the use of aggressive multimodal treatment, the prognosis for patients with t-NENs remains relatively poor. In our cohort, the median OS was 127 months (~10.5 years); however, a considerable variation was observed depending on the treatment intent. Patients who underwent curative-intent treatment achieved a median OS of approximately 170 months, whereas in the palliative group, the median OS was only 33 months. This substantial difference underscores the well-established fact that only early-stage disease—with a resectable tumor—offers a realistic chance for long-term survival.

For comparison, in the study by Filosso et al. (2015), the 5-year OS rate for all patients was 68%, with a median OS of approximately 90 months (7.5 years) [9]. A similar outcome was reported by Zhai et al. (2022), where the 5-year OS reached 66.2% [11]. Our results fall within this range. This indicates that a significant proportion of patients—despite initially successful treatment—eventually experience disease progression and, unfortunately, die as a result. These findings clearly highlight that current therapeutic approaches often fail to achieve durable remission and underscore the urgent need for the development of novel treatment strategies.

This study has several important limitations. First, its retrospective nature inherently introduces the possibility of selection bias and incomplete data collection, particularly in older cases where medical records and staging information were occasionally lacking. Second, the small sample size—reflecting the extreme rarity of t-NENs—limits the statistical power of the analyses and precludes more detailed subgroup comparisons or multivariable modeling.

It should be emphasized that the small sample size significantly limits the statistical power of this study and restricts the ability to draw definitive conclusions regarding survival differences. Our results should be interpreted with caution and not as proof of causality. Observed differences may be influenced by multiple confounding factors, including patient selection, tumor stage, and treatment heterogeneity. Therefore, any statements implying causative treatment effects should be regarded as hypothesis-generating rather than confirmatory.

Additionally, histopathological classifications were retrospectively updated to align with the 2021 WHO criteria without re-review by pathologists. Approximately 15% of patients had NET NOS diagnosis. Three patients in our cohort were classified as having NET NOS. While the 2021 WHO classification does not formally recognize a NOS category for thymic neuroendocrine neoplasms, we adopted this designation descriptively in cases where neuroendocrine differentiation was histologically confirmed, but the available data were insufficient to reliably determine tumor grade or subtype. Importantly, this approach has precedent in the literature. For example, Zhai et al. (2022) [11] used a similar “NET NOS” label when subtype classification was not feasible based on pathology records. This recurring challenge likely reflects the historical lack of uniform diagnostic standards and the inherent complexity of thymic neuroendocrine tumors. As such, the use of a NOS designation in retrospective studies—especially those spanning multiple decades—should be interpreted as a necessary compromise to ensure diagnostic consistency within the constraints of available data. Nonetheless, we acknowledge that the absence of re-review by expert pathologists represents a limitation that may introduce classification bias and should be taken into account when interpreting the results.

Finally, heterogeneity in clinical staging (Masaoka-Koga, TNM, or undocumented) further complicates comparisons across patients and with other published series. A retrospective attempt to reassign stage was not undertaken due to incomplete imaging data and surgical reports in a substantial proportion of cases, particularly those diagnosed more than a decade ago. In the absence of consistent and comprehensive documentation, such restaging would carry a high risk of misclassification. Despite these limitations, our findings align with those of larger multi-institutional studies and provide valuable insight into the real-world management and outcomes of patients with this rare tumor entity.

t-NENs remain malignancies with poor prognosis, characterized by a high propensity for early recurrence and distant metastases despite aggressive multimodal treatment. Complete surgical resection continues to represent the cornerstone of potentially curative therapy—often requiring adjuvant radiotherapy—whereas, in the metastatic setting, the efficacy of currently available systemic treatments remains limited.

Further research—ideally multicenter and prospective—is needed to develop more effective treatment strategies for these rare tumors, particularly regarding systemic therapy in patients with advanced disease.

## 5. Conclusions

Our study confirms the aggressive clinical course of t-NENs, their tendency to recur, and the limited efficacy of current systemic therapies. Complete surgical resection is currently considered the best available option for achieving long-term survival; however, even in such cases, the risk of recurrence remains high. Further research and the development of novel treatment options are essential to improve outcomes in this clinically challenging patient population.

## Figures and Tables

**Figure 1 cancers-17-03932-f001:**
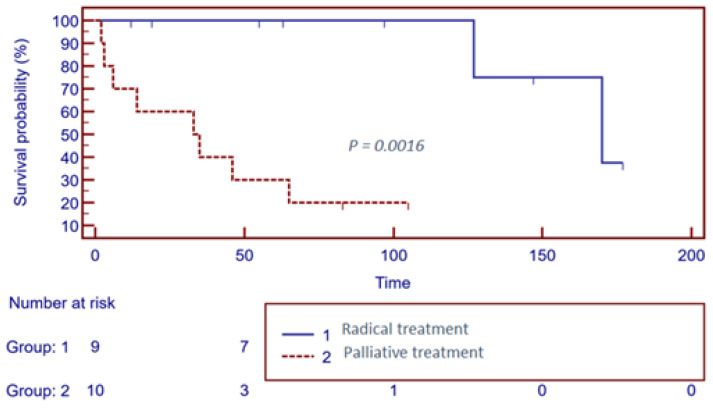
Kaplan–Meier overall survival (OS) curves stratified by treatment intent. The figure shows OS in patients treated with curative intent (Group 1) versus those who received palliative therapy (Group 2). The number of patients at risk at specific time intervals is displayed below the x-axis. Survival curves were compared using the log-rank test.

**Table 1 cancers-17-03932-t001:** Histological subtypes of thymic neuroendocrine neoplasms (t-NEN) in the study cohort.

Histological Subtype	N (%)
Atypical carcinoid	10 (52.6)
LCNEC	6 (31.6)
NET NOS	3 (15.8)

**Abbreviation****s**: LCNEC—large-cell neuroendocrine carcinoma; NET NOS—neuroendocrine tumor not otherwise specified.

**Table 2 cancers-17-03932-t002:** Treatment Modalities in Patients with Thymic Neuroendocrine Neoplasms.

Variable	N (%)
**Primary treatment intent**	
Curative	9 (47.4)
Palliative	10 (52.6)
**Surgical treatment**	
Surgery performed	8 (42.1)
Complete resection (R0)	2 (25.0)
Microscopic residual disease (R1)	6 (75.0)
**Adjuvant therapy after surgery**	
Radiotherapy	7 (36.8)
Chemotherapy (PE regimen)	3 (15.8)
**Palliative systemic treatment**	
Systemic chemotherapy	8 (80.0)
Best supportive care only	2 (20.0)
**Recurrence/progression**	
Recurrence after curative therapy	4 (44.4)
Salvage radical treatment after recurrence	1 (11.1)
Salvage palliative treatment after recurrence	3 (33.3)
Progression after palliative therapy	8 (80.0)
Salvage palliative treatment after progression	7 (70.0)

**Abbreviations**: PE regimen–cisplatin/etoposide.

**Table 3 cancers-17-03932-t003:** Treatment Outcomes in Patients with Thymic Neuroendocrine Neoplasms.

Survival Outcomes	Months [95% CI]
Median OS for all patients	127 [33–170]
Median DFS for curatively treated patients	90 [42–90]
Median OS for curatively treated patients	170 [127–170]
Median PFS for palliatively treated patients	11 [3–105]
Median OS for palliatively treated patients	33 [2–65]

**Abbreviations**: Data are presented as median [95% CI]. CI, confidence interval; OS, overall survival; DFS, disease-free survival; PFS, progression-free survival.

**Table 4 cancers-17-03932-t004:** Treatment characteristics and survival outcomes from retrospective series of surgically treated thymic neuroendocrine tumors.

Author(s)	Total N	N Treated Surgically (%)	R0 Resection (%)	N Receiving Adjuvant Therapy	5-Year OS (%) or Median OS (Months)
Moran & Suster, 2000 [13]	80	74 (92%)	NA	NA	5-year OS: 53%; Median OS: 78 months
Tiffet et al., 2003 [14]	15	15 (100%)	66.7%	12	5-year OS: 53%
Gaur et al., 2010 [12]	160	96 (60%)	NA	35	NA
Cardillo et al., 2012 [6]	20	20 (100%)	95.0%	18	5-year OS: 84%
Crona et al., 2013 [15]	25	14 (56%)	10.7%	8	5-year OS: 79%
Filosso et al., 2015 [9]	205	175 (85%)	57.5%	23	5-year OS: 68%; Median OS: 89.4 months
Zhai et al., 2022 [11]	56	38 (68%)	NA	35	Median OS: 66.1 months
Yliaska et al., 2022 [4]	16	12 (75%)	50.0%	10	5-year OS: 63%; Median OS: 94.0 months
Cheng et al., 2024 [8]	29	13 (45%)	NA	6	Median OS: 66.3 months
Present study 2025	19	8 (42%)	25.0%	8	Median OS: 127 months

**Abbreviations**: OS—overall survival; NA—not available.

## Data Availability

Data may be available upon reasonable request and with permission of the National Research Institute of Oncology, Warsaw, Poland.

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
