# Peer review of "Clinicopathologic Characteristics, Treatment Outcomes, and Survival in Thymic Neuroendocrine Neoplasms (t-NEN): A 25-Year Single-Center Experience"

_cancers, 2025, doi:10.3390/cancers17243932_

Round 1

Reviewer 1 Report

Comments and Suggestions for Authors

The manuscript addresses an important and underexplored topic. Given the rarity of thymic neuroendocrine neoplasms, every institutional experience adds value. The paper is clearly written, methodologically sound, and provides data consistent with existing literature. However, there are several issues that should be addressed before acceptance to strengthen its scientific rigor and clarity.

Major Comments

  1. The cohort of 19 patients is understandable due to disease rarity, but the authors should explicitly acknowledge the limited statistical power and avoid over-interpreting survival comparisons. Statements implying causality (e.g., “radiotherapy improves local control”) should be toned down or referenced appropriately.
  2. The decision to retrospectively update histologic classification to WHO 2021 without re-review by pathologists introduces potential bias. This should be emphasized more strongly in the limitations. Consider providing a supplementary table summarizing original versus updated classifications for transparency.
  3. The lack of a uniform staging system (Masaoka-Koga vs TNM vs none) is a significant limitation. The authors should discuss whether outcomes differed across the subgroups where staging was available and clarify why retrospective restaging was not attempted even approximately.
  4. Specify the number of censored patients and whether survival data were updated for all 19 patients through 2025.
  5. The discussion could better distinguish between data derived from the current cohort and those extrapolated from the literature.
  6. The section on novel therapies (temozolomide, immunotherapy, PRRT) is interesting but reads as a mini-review. Condense it and refocus on how these insights relate to the current findings.
  7. The authors mention two MEN1 cases, but no detailed clinical information is provided. It would be important to clarify: whether these patients exhibited other MEN1-related manifestations (e.g., hyperparathyroidism, pancreatic or pituitary tumors, etc.);  whether the MEN1 diagnosis was genetically confirmed or based solely on clinical criteria; what the clinical course and prognosis were for these patients; and, in case of death, whether it was related to the thymic tumor or to another MEN1-associated manifestation. Sometimes MEN-1 had different clinical course (see also PMID: 36436191)

Minor Comments

  1. The “Simple Summary” is well-written but could highlight key quantitative findings (e.g., median OS, recurrence rate).
  2. Several abbreviations (e.g., PE regimen) should be defined at first mention in the abstract or methods.
  3. Typographical issues: there are several minor formatting inconsistencies (e.g., “largecell” instead of “large-cell,” spaces around parentheses).
  4. Table 4 is valuable but would benefit from a clearer caption and harmonized data formatting (align “median OS” vs “5-year OS”).

Author Response

Response to Reviewer #1

We thank the Reviewer for the thoughtful and constructive comments. Below, we provide a point-by-point response to each comment.

Major Comments

Comment 1: The cohort of 19 patients is understandable due to disease rarity, but the authors should explicitly acknowledge the limited statistical power and avoid over-interpreting survival comparisons. Statements implying causality (e.g., “radiotherapy improves local control”) should be toned down or referenced appropriately.
Response 1: We agree and have revised the Discussion to clearly state that the small sample size significantly limits the statistical power and that observed survival differences are hypothesis-generating rather than confirmatory. Statements implying causality have been appropriately toned down or supported with literature references.

Comment 2: The decision to retrospectively update histologic classification to WHO 2021 without re-review by pathologists introduces potential bias. This should be emphasized more strongly in the limitations. Consider providing a supplementary table summarizing original versus updated classifications for transparency.
Response 2: We acknowledge the reviewer’s concern and have now explicitly emphasized in the revised Discussion that the absence of re-review by experienced pathologists is a potential source of bias. Specifically, we added a sentence stating that this limitation may introduce classification bias and should be considered when interpreting the results.

Comment 3: The lack of a uniform staging system (Masaoka-Koga vs TNM vs none) is a significant limitation. The authors should discuss whether outcomes differed across the subgroups where staging was available and clarify why retrospective restaging was not attempted even approximately.
Response 3: We have expanded the Discussion to address this issue. We explain that retrospective staging was not undertaken due to incomplete imaging and surgical data in many cases, especially older ones, and that attempting it would risk misclassification.

Comment 4: Specify the number of censored patients and whether survival data were updated for all 19 patients through 2025.
Response 4: We confirm that survival data were updated for all 19 patients as of March 31, 2025, with no patients lost to follow-up. This is now clearly stated in the Methods.

Comment 5: The discussion could better distinguish between data derived from the current cohort and those extrapolated from the literature.
Response 5: The Discussion has been revised to consistently distinguish our findings from literature data.

Comment 6: The section on novel therapies (temozolomide, immunotherapy, PRRT) is interesting but reads as a mini-review. Condense it and refocus on how these insights relate to the current findings.
Response 6: This section has been shortened and restructured to emphasize relevance to our findings and observed treatment limitations.

Comment 7: The authors mention two MEN1 cases, but no detailed clinical information is provided. It would be important to clarify: whether these patients exhibited other MEN1-related manifestations (e.g., hyperparathyroidism, pancreatic or pituitary tumors, etc.); whether the MEN1 diagnosis was genetically confirmed or based solely on clinical criteria; what the clinical course and prognosis were for these patients; and, in case of death, whether it was related to the thymic tumor or to another MEN1-associated manifestation.
Response 7: As detailed MEN1 data were not available, we acknowledged this in the manuscript. We expanded the discussion on MEN1 using literature references (including Massironi et al., 2023) to address MEN1 heterogeneity and associated prognosis.

Minor Comments

Comment 1: The “Simple Summary” is well-written but could highlight key quantitative findings (e.g., median OS, recurrence rate).
Response 1: The Simple Summary has been revised to include the median overall survival and recurrence rate.

Comment 2: Several abbreviations (e.g., PE regimen) should be defined at first mention in the abstract or methods.
Response 2: Abbreviations such as “PE regimen” have been defined at first mention in the Abstract and Methods.

Comment 3: Typographical issues: there are several minor formatting inconsistencies (e.g., “largecell” instead of “large-cell,” spaces around parentheses).
Response 3: These typographical inconsistencies have been corrected.

Comment 4: Table 4 is valuable but would benefit from a clearer caption and harmonized data formatting (align “median OS” vs “5-year OS”).
Response 4: Table 4’s caption has been revised for clarity, and formatting has been harmonized to consistently present survival metrics.

We believe these revisions address all concerns and have improved the manuscript. Thank you again for your valuable feedback.

Reviewer 2 Report

Comments and Suggestions for Authors

The manuscript by Piórek et al. is a well written, timely work that provides additional clarity, detail, thoughtful consideration, and recommendations for moving forward with this rare and challenging form of cancer.  I believe this work to be helpful for informing clinicians and patients and/or caregivers needed information for patient advocacy to achieve a quality diagnosis and extend quality of life.  Questions and clarifications will elevate the impact of this work.

Line 61 of the introduction addresses MEN1 hereditary variant of this disease. Information is needed in the introduction and discussion to address oncogene mutational analyse, has this been done in any of the cases? Is it in the disease protocols, if not, why not, if so please detail. Against a landscape of increasing importance of personalized medicine, knowing the tumor driving mutations in the biopsy and germline could be very advantageous; information on the subject would inform the reader if that the possibility could exist.

Line 140 addresses the utilization of cisplatin-etoposide regimen for a treatment, then line 142 addresses PE regimen.  PE is not defined, unless a specialist in the field is reading, there is no clarification what PE means and that it is related to the cisplatin-etoposide regiment.  Please clarify these points in both cases so that it is clear to the reader.

Figure 1 survival curve needs help.  There are a number of questions and confusions that exist based on this figure.  Why are numbers for group 1 and 2 hanging over to additional lines? The legend box appears to be covering data and is out of place, why does group one list of numbers look shorted? What do the numbers for group 1 and 2 mean? No column titles for the data of these two groups.  The caption is embarrassingly simple, please describe more of what is shown if the figure shows as intended. 

Line 250, SCLC is not defined.  The educated reader is left to speculate if this is small cell lung carcinoma, non-specialists will be left to go look up the possible meanings of SCLC.

Author Response

Response to Reviewer #2

We thank the Reviewer for the thoughtful and constructive comments. Below, we provide a point-by-point response to each comment.

Comment 1: Line 61 of the introduction addresses MEN1 hereditary variant of this disease. Information is needed in the introduction and discussion to address oncogene mutational analysis, has this been done in any of the cases? Is it in the disease protocols, if not, why not, if so please detail. Against a landscape of increasing importance of personalized medicine, knowing the tumor driving mutations in the biopsy and germline could be very advantageous; information on the subject would inform the reader if that the possibility could exist.

Response 1: We thank the Reviewer for highlighting this important aspect. In our study, molecular profiling and germline testing were not routinely performed. However, in response to this comment, we have added a paragraph in the Discussion emphasizing the potential role of somatic and germline mutational analysis in the management of t-NENs. We also discuss molecular markers such as MEN1 mutations, MGMT, and PD-L1 expression as potentially informative targets, as reported in recent literature. The new text underscores the relevance of incorporating molecular diagnostics as next-generation sequencing becomes more accessible.

Comment 2: Line 140 addresses the utilization of cisplatin-etoposide regimen for a treatment, then line 142 addresses PE regimen. PE is not defined, unless a specialist in the field is reading, there is no clarification what PE means and that it is related to the cisplatin-etoposide regiment. Please clarify these points in both cases so that it is clear to the reader.

Response 2: We appreciate the Reviewer’s suggestion. We have now clarified and defined “PE regimen” as “cisplatin-etoposide”.

Comment 3: Figure 1 survival curve needs help. There are a number of questions and confusions that exist based on this figure. Why are numbers for group 1 and 2 hanging over to additional lines? The legend box appears to be covering data and is out of place, why does group one list of numbers look shorted? What do the numbers for group 1 and 2 mean? No column titles for the data of these two groups. The caption is embarrassingly simple, please describe more of what is shown if the figure shows as intended.

Response 3: Thank you for this detailed feedback. Due to limitations in our source file format, we were unable to fully reconstruct all graphical elements, but we ensured that all relevant survival data, group labels, and patient numbers at risk are now clearly visible. The figure caption has also been expanded to provide a more comprehensive description of the contents and statistical methods used.

Comment 4: Line 250, SCLC is not defined. The educated reader is left to speculate if this is small cell lung carcinoma, non-specialists will be left to go look up the possible meanings of SCLC.

Response 4: Thank you for pointing this out. We have now defined “SCLC” as “small cell lung carcinoma” at its first mention in the manuscript to ensure clarity for all readers.

Round 2

Reviewer 1 Report

Comments and Suggestions for Authors

Thank you for your careful revisions. All major concerns have been satisfactorily addressed, and the manuscript has clearly improved.

Remaining issues are limited to minor editorial refinements:

  1. A few small editorial repetitions and redundant sentences remain in the Discussion (likely due to revisions without detailed language editing); we recommend minor polishing for clarity.
  2.  Ensure consistent formatting of percentages, confidence intervals, and abbreviations throughout the manuscript.